# A general strategy to construct small molecule biosensors in eukaryotes

**Justin Feng[1,2†], Benjamin W Jester[3,4†], Christine E Tinberg[5†], Daniel J Mandell[2,6*†], Mauricio S Antunes[7], Raj Chari[2], Kevin J Morey[7], Xavier Rios[2], June I Medford[7], George M Church[2,6*], Stanley Fields[3,4,8*], David Baker[4,5*]**

[1]Program in Biological and Biomedical Sciences, Harvard Medical School, Boston, United States; [2]Department of Genetics, Harvard Medical School, Boston, United States; [3]Department of Genome Sciences, University of Washington, Seattle, United States; [4]Howard Hughes Medical Institute, University of Washington, Seattle, United States; [5]Department of Biochemistry, University of Washington, Seattle, United States; [6]Wyss Institute for Biologically Inspired Engineering, Harvard University, Boston, United States; [7]Department of Biology, Colorado State University, Fort Collins, United States; [8]Department of Medicine, University of Washington, Seattle, United States

**Abstract** Biosensors for small molecules can be used in applications that range from metabolic engineering to orthogonal control of transcription. Here, we produce biosensors based on a ligand-binding domain (LBD) by using a method that, in principle, can be applied to any target molecule. The LBD is fused to either a fluorescent protein or a transcriptional activator and is destabilized by mutation such that the fusion accumulates only in cells containing the target ligand. We illustrate the power of this method by developing biosensors for digoxin and progesterone. Addition of ligand to yeast, mammalian, or plant cells expressing a biosensor activates transcription with a dynamic range of up to ~100-fold. We use the biosensors to improve the biotransformation of pregnenolone to progesterone in yeast and to regulate CRISPR activity in mammalian cells. This work provides a general methodology to develop biosensors for a broad range of molecules in eukaryotes.

**\*For correspondence:**
daniel_mandell@hms.harvard.edu (DJM); gchurch@genetics.med. harvard.edu (GMC); fields@uw. edu (SF); dabaker@uw.edu (DB)

[†]These authors contributed equally to this work

## Introduction

Biosensors capable of sensing and responding to small molecules in vivo have wide-ranging applications in biological research and biotechnology, including metabolic pathway regulation (*Zhang et al., 2012*), biosynthetic pathway optimization (*Raman et al., 2014*; *Tang and Cirino, 2011*), metabolite concentration measurement and imaging (*Paige et al., 2012*), environmental toxin detection (*Gil et al., 2000*), and small molecule-triggered therapeutic response (*Ye et al., 2013*). Despite such broad utility, no single strategy for the construction of biosensors has proven sufficiently generalizable to gain widespread use. Current methods typically couple binding of a small molecule to a single output signal, and use a limited repertoire of natural protein- (*Tang et al., 2013*) or nucleic acid aptamer-binding (*Yang et al., 2013*) domains, which narrows the scope of small molecules that can be detected. A general solution to small molecule biosensing should be adaptable to a range of small molecules and responses.

A promising approach to biosensor design in eukaryotes uses conditionally stable ligand-binding domains (LBDs) (*Banaszynski et al., 2006*; *Tucker and Fields, 2001*). In the absence of a cognate ligand, these proteins are degraded by the ubiquitin proteasome system (*Egeler et al., 2011*). Binding of the ligand stabilizes the LBD and prevents its degradation. Fusing the destabilized LBD to a

**eLife digest** Small molecules play essential roles in organisms, and so methods to sense these molecules within living cells could have wide-ranging uses in both biology and biotechnology. However, current methods for making new "biosensors" are limited and only a narrow range of small molecules can be detected.

One approach to biosensor design in yeast and other eukaryotic organisms uses proteins called ligand-binding domains, which bind to small molecules. Here, Feng, Jester, Tinberg, Mandell et al. have developed a new method to make biosensors from ligand-binding domains that could, in principle, be applied to any target small molecule. The new method involves taking a ligand-binding domain that is either engineered or occurs in nature and linking it to something that can be readily detected, such as a protein that fluoresces or that controls gene expression. This combined biosensor protein is then engineered, via mutations, such that it is unstable unless it binds to the small molecule. This means that, in the absence of the small molecule, these proteins are destroyed inside living cells. However, the binding of a target molecule to one of these proteins protects it from degradation, which allows the signal to be detected.

Feng, Jester, Tinberg, Mandell et al. use this method to create biosensors for a human hormone called progesterone and a drug called digoxin, which is used to treat heart disease. Further experiments used the biosensors to optimize the production of progesterone in yeast and to regulate the activity of a gene editing protein called Cas9 in human cells. The biosensors can be also used to produce long-term environmental sensors in plant cells.

This approach makes it possible to produce a wide variety of biosensors for different organisms. The next step is to continue to explore the ability of various proteins to be converted into biosensors, and to find out how easy it is to transfer a biosensor produced in one species to another.

suitable reporter protein, such as an enzyme, fluorescent protein, or transcription factor, renders the fusion conditionally stable and generates sensor response (*Figure 1a*). Naturally-occurring LBDs can be engineered to be conditionally stable (*Banaszynski et al., 2006*; *Miyazaki et al., 2012*; *Iwamoto et al., 2010*), making it possible in principle to convert them into biosensors for target ligands. Designed LBDs can also be used, especially in cases for which natural binding proteins do not exist or lack sufficient specificity or bioorthogonality.

Here, we convert a single designed LBD scaffold into multiple highly specific biosensors for the clinically relevant steroids digoxin and progesterone. We engineer LBDs fused with fluorescent reporters to be conditionally stable in the budding yeast *Saccharomyces cerevisiae*. Attaching these conditionally-stabilized LBDs to transcription factors (TFs) yields biosensors that respond to their target ligands with greater signal induction than observed with fusions to fluorescent proteins. We use TF-biosensors to improve the biosynthetic yield of progesterone in yeast. The biosensors retain function when ported directly into mammalian cells, with up to 100-fold activation over background, allowing us to develop a method for tight control of CRISPR/Cas9 genome editing. The biosensors also show up to 50-fold activation by ligand in *Arabidopsis thaliana*. The method presented here enables the rapid development of eukaryotic biosensors from natural and designed binding domains.

## Results

### Fluorescent biosensors built from engineered LBDs

LBDs intended for biosensor development should recognize their targets with high affinity and specificity. We began with the computationally-designed binding domain DIG10.3 (*Tinberg et al., 2013*), hereafter $DIG_0$, which binds the plant steroid glycoside digoxin and its aglycone digoxigenin with picomolar affinities. Introduction of three rationally-designed binding site mutations into $DIG_0$ resulted in a progesterone binder ($PRO_0$) with nanomolar affinity (*Tinberg et al., 2013*). We constructed genetic fusions of $DIG_0$ and $PRO_0$ to a yeast-enhanced GFP (yEGFP, LBD-biosensors $DIG_0$-

GFP and PRO$_0$-GFP) and constitutively expressed them in *S. cerevisiae* (*Supplementary file 1*). The fusions showed little change in fluorescence in response to digoxin or progesterone, respectively (*Figure 1b,c* and *Figure 1—figure supplement 1*). Work by Wandless and co-workers has shown that mutagenesis of LBDs can be used to identify variants that are stable only in the presence of a target ligand (*Banaszynski et al., 2006*). We randomly mutagenized the LBDs of DIG$_0$-GFP and PRO$_0$-GFP by error-prone PCR and subjected libraries of $10^5$ (*Gil et al., 2000*) integrants to multiple rounds of FACS, sorting alternately for high fluorescence in the presence of the ligand and low fluorescence in its absence. We isolated LBD variants having greater than 5-fold activation by cognate ligand (*Figure 1b,c* and *Figure 1—figure supplement 1*). By making additional variants that contain only one of the up to four mutations found in the progesterone biosensors, we showed that some mutations are additive, while others predominately contribute to sensitivity (*Figure 1—figure supplement 2a*). Many of the conditionally-destabilizing mutations identified in DIG$_0$ involve residues that participate in key dimer interface interactions (*Figure 1d*). The conditionally-destabilizing mutations of PRO$_0$ are located throughout the protein (*Figure 1—figure supplement 2b–d*); the DIG$_0$ interface mutations also rendered PRO$_0$-GFP conditionally stable on binding progesterone (*Figure 1—figure supplement 2e*).

## TF-biosensors amplify ligand-dependent responses

To improve the dynamic range and utility of the biosensors, we built conditionally-stable LBD-transcription factor fusions (TF-biosensors) by placing an LBD between an N-terminal DNA-binding domain (DBD) and a C-terminal transcriptional activation domain (TAD, *Figure 2a*). The use of TFs serves to amplify biosensor response and allows for ligand-dependent control of gene expression (*Shoulders et al., 2013*; *Beerli et al., 2000*; *Louvion et al., 1993*). Our initial constructs used the DBD of Gal4, the destabilized LBD mutant DIG$_1$ (E83V), and either VP16 or VP64 as a TAD to drive the expression of yEGFP under the control of a *GAL1* promoter. The dynamic range of TF-biosensor activity was maximal when the biosensor was expressed using a weak promoter and weak activation domain, because of lower yEGFP expression in the absence of ligand (*Figure 2—figure supplement 1a,b*).

We chose Gal4-DIG$_1$-VP16 (hereafter G-DIG$_1$-V) for further TF-biosensor development because it has both a large dynamic range and maximal activation by ligand. A FACS-based screen of an error-prone PCR library of G-DIG$_0$-V, G-DIG$_1$-V, and G-DIG$_2$-V variants identified mutations L77F and R60S in the Gal4 dimer interface (hereafter G$_{L77F}$, G$_{R60S}$) that further increased TF-biosensor response by lowering background activity in the absence of ligand (*Figure 2b* and *Figure 2—figure supplement 1c*). Although these Gal4 mutations were identified by screening the libraries of digoxin-dependent TF-biosensors, they also increased progesterone-dependent activation of the G-PRO-V series of biosensors, indicating a shared mechanism of conditional stability in both systems (*Figure 2—figure supplement 1d*). Combining mutations in Gal4 and DIG$_0$ or PRO$_0$ led to activations of up to 60-fold by cognate ligand, a ten-fold improvement over the most responsive LBD-biosensors (*Figure 2c,d* and *Figure 2—figure supplement 2a*) and a dynamic range that has been challenging to achieve with stability-based biosensors in yeast (*Rakhit et al., 2011*). The TF-biosensors were also rapidly activated, showing a five-fold increase in signal after 1 hr of incubation with ligand and full activation after ~14 hr (*Figure 2e,f* and *Figure 2—figure supplement 2b*). In contrast to the LBD-biosensors, the TF-biosensors exhibited a broad range of fluorescence levels across single cells, as well as a population of nonfluorescent cells in the presence of ligand (*Figure 2—figure supplement 2*). We used FACS to isolate cells from the nonfluorescent population and found those cells to be inviable, possibly indicating plasmid loss or toxicity from biosensor activation.

Upon withdrawal of ligand, strains expressing TF-biosensors rapidly exhibited reduction in signal, reaching half of their maximum yEGFP fluorescence after approximately 5 hr and nearly undetectable fluorescence after 10–15 hr (*Figure 2g,h*). The response of the TF-biosensors to the withdrawal of ligand is likely much faster than observed by fluorescence, as the reduction in fluorescence signal is dependent on both the degradation of the TF-biosensors as well as the degradation and dilution of previously expressed yEGFP.

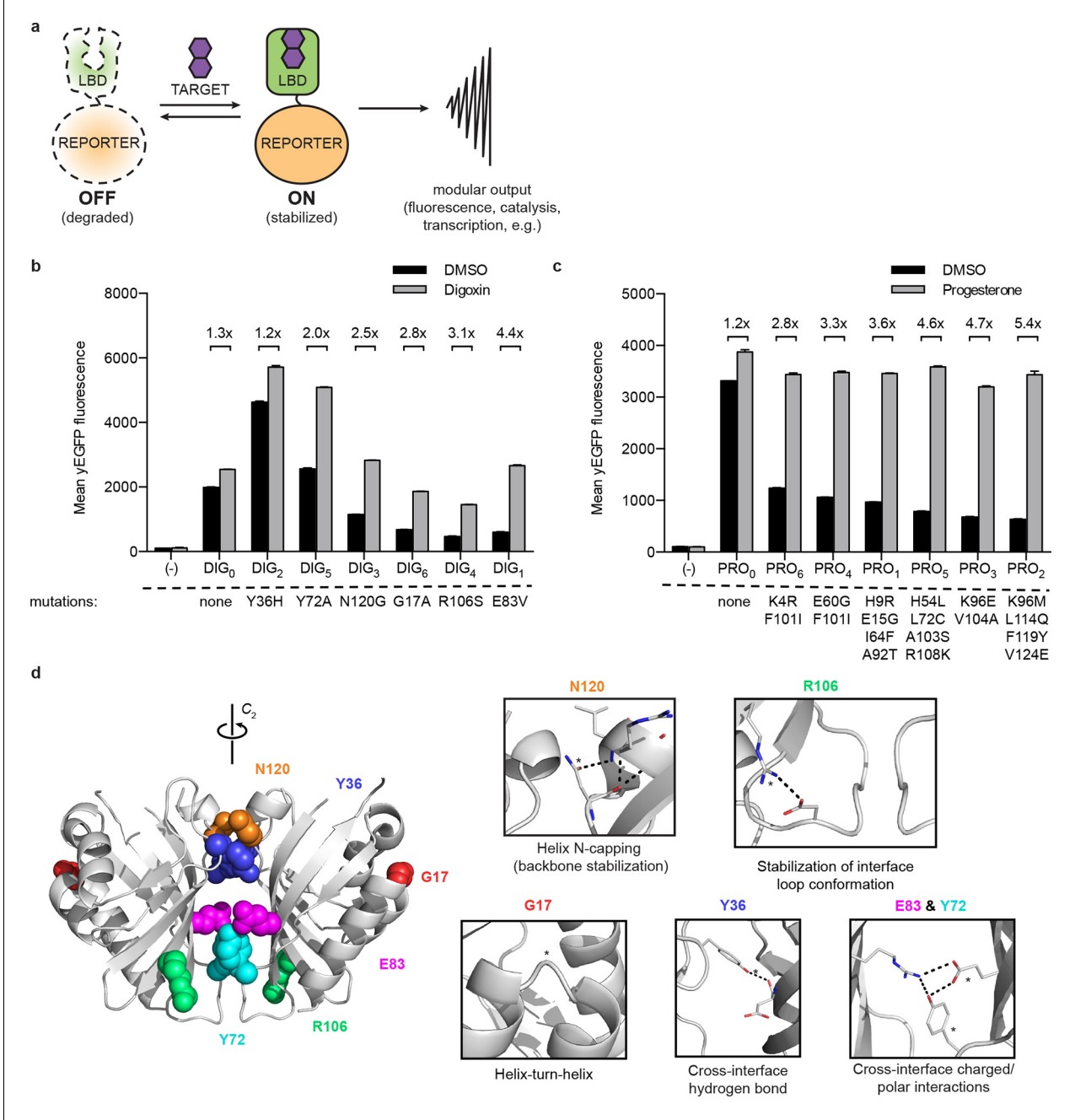

**Figure 1.** A general method for construction of biosensors for small molecules. (a) Modular biosensor construction from a conditionally destabilized LBD and a genetically fused reporter. The reporter is degraded in the absence but not in the presence of the target small molecule. (b) yEGFP fluorescence of digoxin LBD-GFP biosensors upon addition of 250 μM digoxin or DMSO vehicle. (c) yEGFP fluorescence of progesterone LBD-GFP biosensors upon addition of 50 μM progesterone or DMSO vehicle. (d) Positions of conditionally destabilizing mutations of $DIG_0$ mapped to the crystal structure of the digoxin LBD (PDB ID 4J9A). Residues are shown as colored spheres and key interactions highlighted in insets. In b-c, fold activation is shown above brackets, (-) indicates cells lacking biosensor constructs, and error bars represent the standard error of the mean (s.e.m.) of three biological replicates.

The following figure supplements are available for figure 1:

**Figure supplement 1.** Population responses to cognate ligand for cells bearing LBD-biosensors.

**Figure supplement 2.** Characterization of mutations conferring progesterone-dependent stability.

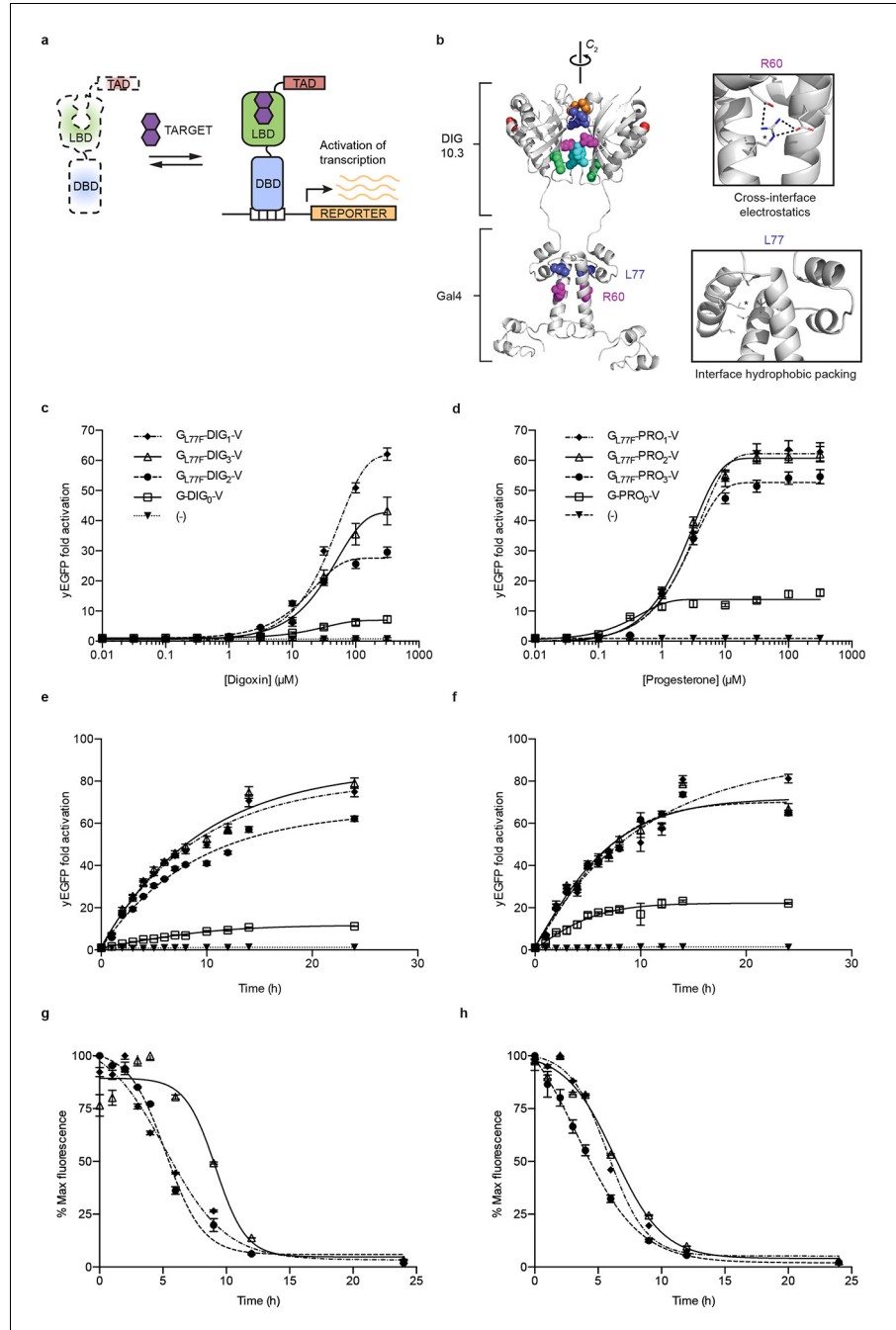

**Figure 2.** Ligand-dependent transcriptional activation. (**a**) TF-biosensor construction from a conditionally destabilized LBD, a DNA-binding domain and a transcriptional activation (TAD) domain. (**b**) Positions of conditionally destabilizing mutations of Gal4 mapped to a computational model of Gal4-DIG$_0$ homodimer. Residues are shown as colored spheres and key interactions are highlighted in insets. The TAD is not shown. (**c**) Concentration dependence of response to digoxin for digoxin TF-biosensors driving yEGFP expression. (**d**) Concentration dependence of response to progesterone for progesterone TF-biosensors driving yEGFP expression. (**e**) Time dependence of response to 250 µM digoxin for digoxin TF-biosensors. (**f**) Time dependence of response to 50 µM progesterone for progesterone TF-biosensors. (**g**) Time-dependent response to withdrawal of 250 µM digoxin for digoxin TF-biosensors. (**h**) Time-dependent response to withdrawal of 50 µM progesterone for progesterone TF-biosensors. In **c-f**, (-) indicates cells lacking biosensor plasmids and error bars represent s.e.m. of three biological replicates. Marker symbols in **e** and **g** are the same as in **c**. Marker symbols in **f** and **h** are the same as in d.

*Figure 2 continued on next page*

*Figure 2 continued*

The following figure supplements are available for figure 2:

**Figure supplement 1.** Improvements to TF-biosensor response.

**Figure supplement 2.** Population responses to cognate ligand for cells bearing TF-biosensors.

## TF-biosensors are tunable and modular

An attractive feature of the TF-biosensors is that the constituent parts – the DBD/promoter pair, the LBD, the TAD, the reporter, and the yeast strain – are modular, such that the system can be modified for additional applications. To demonstrate tunability, we replaced the DBD of $G$-$DIG_1$-$V$ with the bacterial repressor LexA and replaced the Gal4 DNA-binding sites in the *GAL1* promoter with those for LexA. LexA-based TF-biosensors with $DIG_1$ and a weak TAD (B42) showed a strong response to digoxin (nearly 40-fold) only when the promoter-driving reporter expression contained LexA-binding sites (*Figure 3a*). These results demonstrate that the biosensors can function with different combinations of DBDs and TADs, which could produce diverse behaviors and permit their use in eukaryotes requiring different promoters. Furthermore, the reporter gene can be swapped with an auxotrophic marker gene to enable growth selections. The TF-biosensors drove the expression of the *HIS3* reporter most effectively when steroid was added to the growth media, as assessed by the growth of a histidine auxotrophic strain in media lacking histidine (*Figure 3b,d*). Fusion of the Matα2 degron to the biosensor improved dynamic range by reducing the growth of yeast in the absence of ligand. Finally, the yeast strain could be modified to improve biosensor sensitivity toward target ligands by the deletion of the gene for a multidrug efflux pump (*Ernst et al., 2005*), thereby increasing ligand retention (*Figure 3c–d*).

## TF-biosensors enable a selection in yeast to improve the bioproduction of a small molecule

Improving bioproduction requires the ability to detect how modifications to the regulation and composition of production pathways affect product titers. Current product detection methods such as mass spectrometry or colorimetric assays are low-throughput and are not scalable or generalizable. LBD- and TF-biosensors could be coupled with fluorescent reporters to enable high throughput library screening or to selectable genes to permit rapid evolution of biosynthetic pathways (*Tang and Cirino, 2011*; *Dietrich et al., 2010*; *Chou and Keasling, 2013*). Yeast-based platforms have been developed for the biosynthesis of pharmaceutically relevant steroids, such as progesterone and hydrocortisone (*Duport et al., 1998*; *Szczebara et al., 2003*). A key step in the production of both steroids is the conversion of pregnenolone to progesterone by the enzyme 3β-hydroxysteroid dehydrogenase (3β-HSD). We aimed to use a progesterone biosensor to detect and improve this transformation. An important feature of biosensors intended for pathway engineering is their ability to detect a product with minimal activation by substrate or other related chemicals. TF-biosensors built from $PRO_1$ showed the greatest dynamic range and selectivity for progesterone over pregnenolone when driving yEGFP expression or when coupled with a *HIS3* reporter assay (*Figure 4a,b* and *Figure 4—figure supplement 1a*). We investigated whether this sensor could be used to detect the in vivo conversion of pregnenolone to progesterone by episomally-expressed 3β-HSD (*Figure 4c*). Using $G_{L77F}$-$PRO_1$-$V$ driving a yEGFP reporter, we could detect progesterone production, with biosensor response greatest when 3β-HSD was expressed from a high copy number plasmid and from a strong promoter (*Figure 4d*).

We then sought to use the biosensor to improve this enzymatic transformation. To select for improved progesterone production, we required a growth assay in which wild-type 3β-HSD could no longer complement histidine auxotrophy when the yeast were grown on plates supplemented with pregnenolone. To this end, the selection stringency was tuned by adding the His3 inhibitor 3-aminotriazole (*Figure 4—figure supplement 1e*). We mutagenized the 3β-HSD coding sequence using error-prone PCR and screened colonies that survived the *HIS3* selection for their yEGFP activation by pregnenolone. By transforming evolved 3β-HSD mutations into a fresh host background, we showed that the mutations in the enzyme, and not off-target plasmid or host escape mutations,

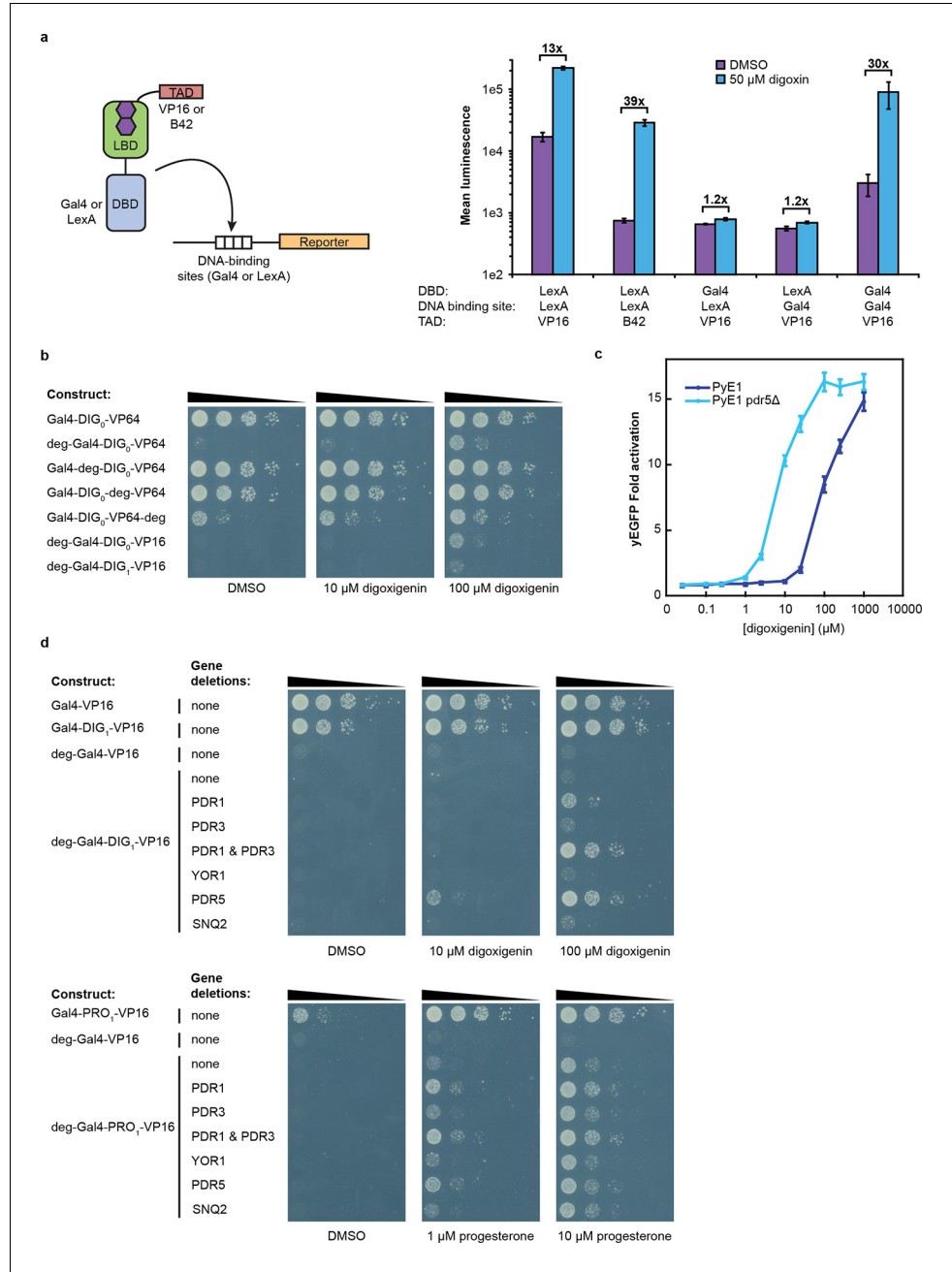

**Figure 3.** Tuning TF-biosensors for different contexts. (**a**) The TAD and DBD of the TF-biosensor and the corresponding binding site for the DBD in the reporter promoter can be swapped for a different application. Expression of a plasmid-borne luciferase reporter was driven by TF-biosensors containing either a LexA or Gal4 DBD and either a VP16 or B42 TAD. Promoters for the reporter contained DNA-binding sites for either Gal4 or LexA. (**b**) TF-biosensors were transformed into the yeast strain PJ69-4a and tested for growth on this minimal media containing 1 mM 3-aminotriazole (3-AT) and the indicated steroid. To determine the effect of including an additional destabilization domain, the degron from Matα2 was cloned into one of four positions. (**c**) G-DIG$_1$-V biosensor response to digoxigenin in yEGFP reporter strain PyE1 either with or without a deletion to the ORF of *PDR5*. (**d**) Ligand and TF-biosensor-dependent growth on this media in yeast strains containing deleted ORFs for efflux-related transcription factors (*PDR1* and *PDR3*) or ABC transporter proteins (*YOR1, PDR5, SNQ2*). In **a** and **c**, error bars represent s.e.m. of three biological replicates.

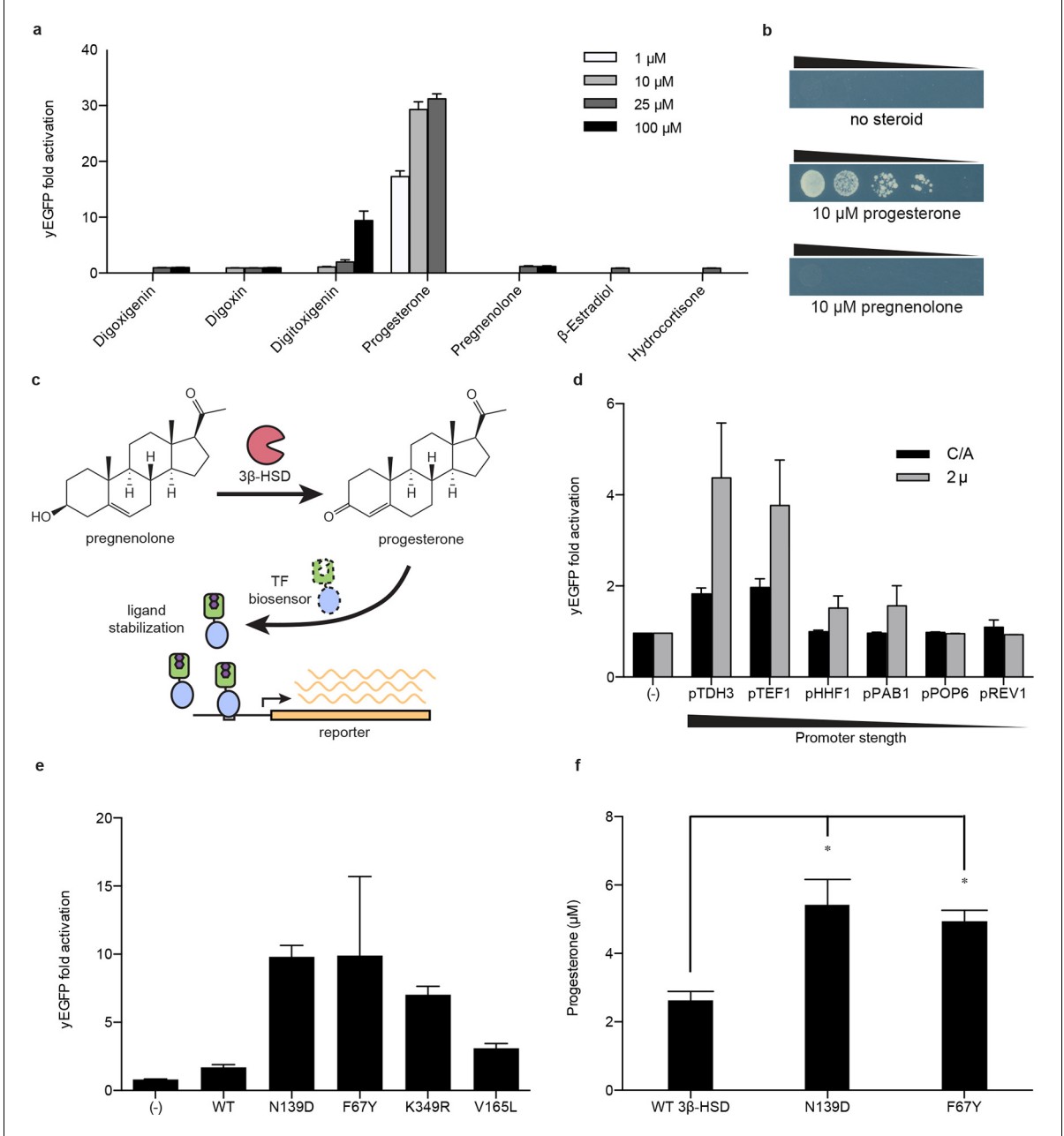

**Figure 4.** Application of biosensors to metabolic engineering in yeast. (a) Fold activation of $G_{L77F}$-$PRO_1$-V by a panel of steroids in yEGFP reporter strain PyE1. Data are represented as mean ± SEM. (b) Growth of degron-G-$PRO_1$-V in *HIS3* reporter strain PJ69-4a is stimulated by progesterone but not pregnenolone. (c) Schematic for directed evolution of 3β-HSD using TF-biosensors for the conversion of pregnenolone to progesterone. (d) Fold activation of $G_{L77F}$-$PRO_1$-V by a panel of plasmids expressing wild-type 3β-HSD under varying promoter strengths in yEGFP reporter strain PyE1 when incubated in 50 μM pregnenolone. Data for plasmids containing CEN/ARS and 2 μ (2 micron) origins are shown. (e) Fold activation of $G_{L77F}$-$PRO_1$-V by a panel of evolved 3β-HSD mutants expressed under the *TDH3* promoter on a CEN/ARS plasmid and incubated in 50 μM pregnenolone. (f) Progesterone titer in 1 OD of cells produced by strains expressing 3β-HSD mutants. Progesterone became toxic at levels of 100 μM and above, leading to substantial cell death. β-estradiol and hydrocortisone were not soluble in yeast growth media at levels above 25 μM. In a and d-f, data are presented as mean ± s.e.m. of three biological replicates. In d and e, (-) indicates cells lacking 3β-HSD. *indicates significance with a threshold of $p < 0.05$ using 2-tailed Student's t-test.

The following figure supplement is available for figure 4:

**Figure supplement 1.** Specificity of PRO biosensors enables selection for auxotrophy complementation.

were responsible for increased biosensor response (*Figure 4e*). Two of the mutants, 3β-HSD N139D and 3β-HSD F67Y, were assayed for progesterone production using gas chromatography and mass spectrometry and were found to produce two-fold more progesterone per OD than cells bearing the wild-type enzyme (*Figure 4f*).

## Yeast-derived biosensors port directly to mammalian cells and can be used to tightly regulate CRISPR/Cas9 genome editing

Yeast is an attractive platform for engineering in vivo biosensors because of its rapid doubling time and tractable genetics. If yeast-derived biosensors function in more complex eukaryotes, the design-build-test cycle in those organisms could be rapidly accelerated. We first assessed the portability of yeast TF-biosensors to mammalian cells. Single constructs containing digoxin and progesterone TF-biosensors with the greatest dynamic ranges (without codon optimization) were stably integrated into human K562 cells using PiggyBac transposition. We characterized the dynamics of the TF-biosensors in human cells by dose response and time course assays similar to the yeast experiments (*Figure 5a–d*). As with yeast, the human cells demonstrated greater sensitivity to digoxin, with fluorescence activation increasing up to 100 nM of cognate ligand for digoxin biosensors and 1 mM for progesterone biosensors. We observed >100-fold activation for the most sensitive progesterone biosensor $G_{L77F}$-PRO$_1$-V. The increase in mammalian dynamic range over yeast may arise from more aggressive degradation of destabilized biosensors or greater accumulation of target-stabilized biosensors or reporters resulting from larger cell sizes and slower doubling times. The time course data show that fluorescence increased four-fold within 4 hr of target introduction and rose logarithmically for 24–48 hr.

We next assessed whether these biosensors could drive more complex mammalian phenotypes. The CRISPR/Cas9 system has proved to be an invaluable tool for genome editing (*Mali et al., 2013*; *DiCarlo et al., 2013*; *Gratz et al., 2013*; *Hwang, 2013*). Despite the high programmability and specificity of Cas9-mediated gene editing achieved to date, unchecked Cas9 activity can lead to off-target mutations and cytotoxicity (*Fu et al., 2013*; *Mali et al., 2013*; *Pattanayak et al., 2013*). Further, it may be desirable to tightly regulate Cas9 activity such that gene editing occurs only under defined conditions. To facilitate inducible gene editing, we fused human codon-optimized versions of the DIG$_3$ and PRO$_1$ LBDs to the N-terminus of Cas9 from *S. pyogenes*. We integrated this construct into a reporter cell line containing an EGFP variant with a premature stop codon that renders it non-functional. Upon separate stable integration of the DIG-Cas9 and PRO-Cas9 fusions, we transfected a guide RNA targeting the premature stop codon as well as a donor oligonucleotide containing the sequence to restore EGFP activity *via* homologous recombination. After a 48-hr incubation period, we observed an ~18-fold increase in GFP positive cells with digoxigenin relative to the mock control (*Figure 5e*).

## Environmental detection in the plant *Arabidopsis thaliana*

To assess generalizability of these sensors to multicellular organisms, we engineered G-DIG$_1$-V to function as an environmental biosensor in plants. The DIG$_1$ sequence was codon optimized for expression in *Arabidopsis thaliana*. We tested biosensor fusions to two different degrons, Matα2 from yeast and DREB2a from *Arabidopsis* (*Sakuma et al., 2006*), and we used the VP16 and VP64 variants as the TAD. We initially tested the G-DIG$_1$-TAD variants with a transient expression assay using *Arabidopsis* protoplasts and a reporter gene consisting of firefly luciferase under the control of a Gal4-activated plant promoter (*pUAS::Luc*). The biosensor containing the Matα2 degron and VP16 TAD showed the highest fold activation of luciferase in the presence of digoxigenin (*Figure 6—figure supplement 1a*). We next inserted the genes encoding G-DIG$_1$-V-Matα2 and the Gal4-activated *pUAS::Luc* into a plant transformation vector and stably transformed them into *Arabidopsis* plants. Primary transgenic plants were screened in vivo for digoxigenin-dependent luciferase production, and responsive plants were allowed to set seed for further testing. Second generation transgenic plants (T$_1$, heterozygous) were tested for digoxin- or digoxigenin-dependent induction of luciferase expression. After 42 hr, we observed 30-50-fold induction of luciferase activity in digoxin-treated plants compared to the uninduced control (*Figure 6*). Both digoxin and digoxigenin were capable of inducing the biosensor. Digoxigenin-dependent luciferase induction was observed in multiple independent transgenic T$_1$ lines (*Figure 6—figure supplement 1b*), and a rising dose response

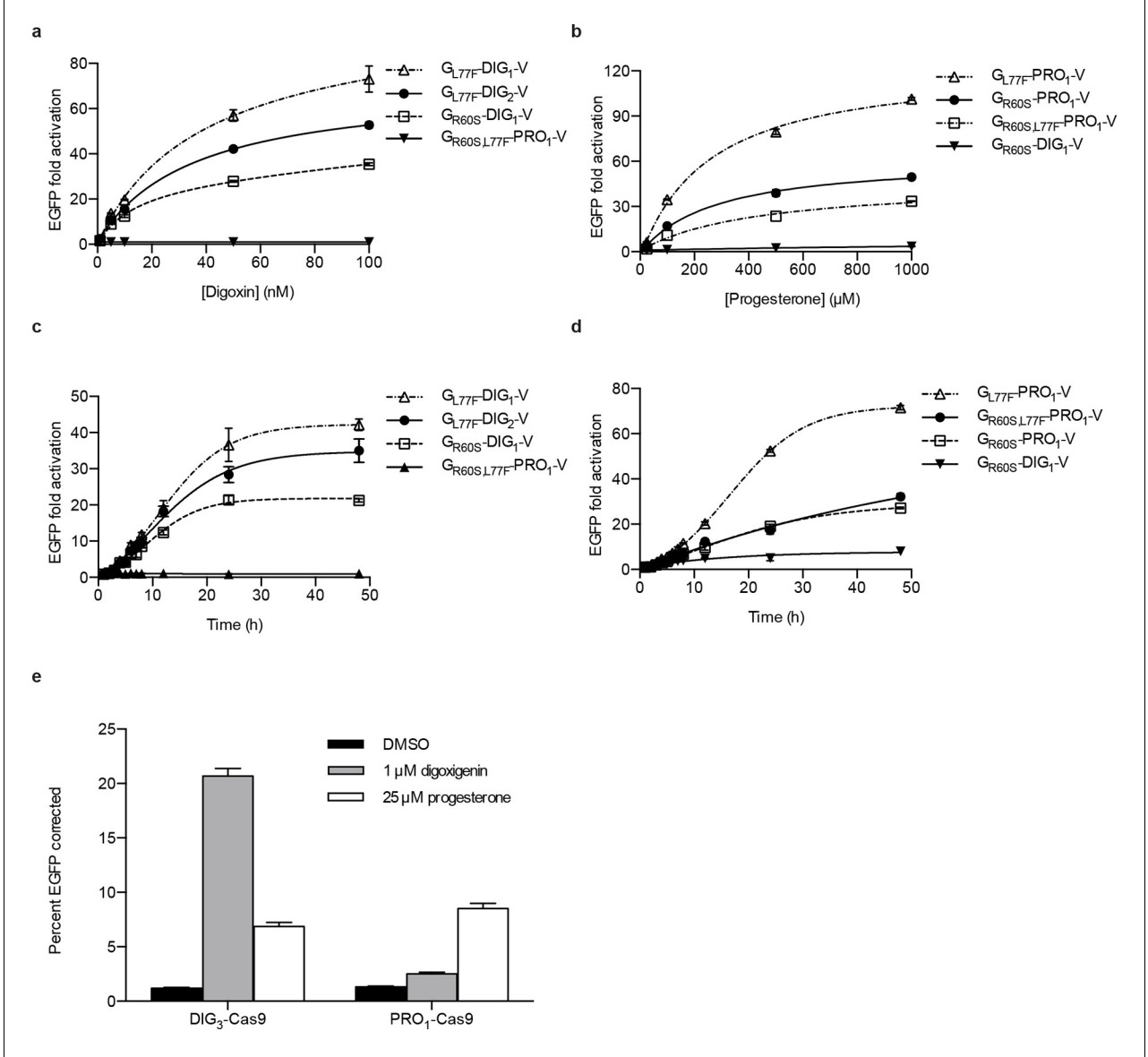

**Figure 5.** Activation of biosensors in mammalian cells and regulation of CRISPR/Cas9 activity. (**a**) Concentration dependence of response to digoxin for constructs containing digoxin TF-biosensors and Gal4 UAS-E1b-EGFP reporter individually integrated into K562 cells. $G_{R60S,L77F}$-$PRO_1$-V serves as a digoxin insensitive control. (**b**) Concentration dependence of response to progesterone for constructs containing progesterone TF-biosensors and Gal4 UAS-E1b-EGFP reporter individually integrated into K562 cells. $G_{R60S}$-$DIG_1$-V serves as a progesterone insensitive control. (**c**) Time dependence of response to 100 nM digoxin for constructs containing digoxin TF-biosensors and Gal4 UAS-E1b-EGFP reporter individually integrated into K562 cells. $G_{R60S,L77F}$-$PRO_1$-V serves as a digoxin insensitive control. (**d**) Time dependence of response to 25 μM progesterone for constructs containing progesterone TF-biosensors and Gal4 UAS-E1b-EGFP reporter individually integrated into K562 cells. $G_{R60S}$-$DIG_1$-V serves as a progesterone insensitive control. (**e**) $DIG_3$ and $PRO_1$ fused to the N-terminus of *S. pyogenes* Cas9 were integrated into a K562 cell line containing a broken EGFP. EGFP function is restored upon transfection of a guide RNA and donor oligonucleotide with matching sequence in the presence of active Cas9. The data are presented as mean fluorescence ± s.e.m. of three biological replicates.

to digoxigenin was observed in the transgenic plants (*Figure 6—figure supplement 1c*). The specificity of the digoxigenin biosensor in plants parallels that in yeast cells (*Figure 6—figure supplement 1d*).

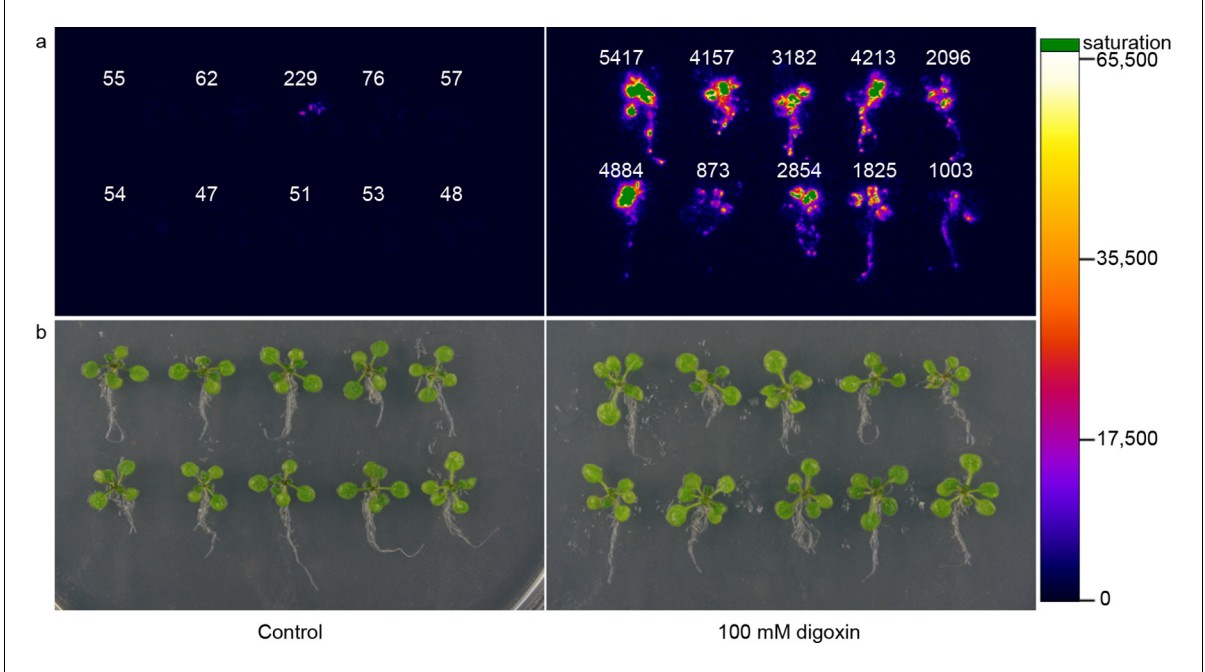

**Figure 6.** Application of biosensors in plants. (**a**) Activation of luciferase expression in transgenic *Arabidopsis* plants containing the G-DIG$_1$-V biosensor in the absence (left) or presence (right) of 100 µM digoxin. Luciferase expression levels are false colored according to scale to the right. Relative luciferase units corresponding to 1 min of image pixel integration (to avoid saturating pixels) are shown above each individual plant. (**b**) Brightfield image of plants shown in **a**.

The following figure supplement is available for figure 6:

**Figure supplement 1.** Characterization of DIG biosensor in plants.

## Discussion

In vivo biosensors for small molecules enable the regulation and detection of cellular responses to endogenous metabolites and exogenous chemicals. Here, we show that LBDs can be conditionally destabilized to create biosensors that function in yeast, mammalian cells, and plants, and we demonstrate the use of these biosensors in metabolic engineering and genome editing applications. While this method requires a high-affinity ligand-binding domain as a starting point, nearly all small molecules of interest have a natural protein interactor. Furthermore, the use of de novo-designed binders opens the possibility of generating biosensors for ligands with unsuitable or unknown binding proteins. By incorporating standard mutagenesis and screening, our method constitutes a simple platform for sensor development that can be applied to many areas of biotechnology. These sensors act either at the level of post-translational control over protein function or at the level of transcription (*Figure 7a*), and they can be tuned by altering any of their components (*Figure 7b*) or by modifying efflux of the target ligand in the host organism. These tunable features should make the biosensors useful in many different cellular and environmental contexts.

Our results suggest a general mechanism of conditional stabilization for LBDs, allowing the rational development of biosensors for other targets. Furthermore, the portability of the mutations we identified suggests a structural basis for conferring conditional stability to the DIG$_0$ LBD scaffold. Both the DIG$_0$ LBD and Gal4 are homodimers, and the majority of the conditionally-stabilizing mutations are located at the dimer interfaces. A computational model of the Gal4-DIG$_0$ complex indicates that the orientation of the two domains allows a homodimeric fusion to form (*Figure 2b*). These results suggest an allosteric interplay between ligand binding and dimer formation: weakening of the dimer interface, in either the DIG$_0$ or the Gal4 domain, is compensated by ligand binding. This LBD scaffold is derived from a member of the nuclear transport factor 2 family, a fold class that typically has a large dimer interface (~1200 Å$^2$) that facilitates the large and open ligand-binding site

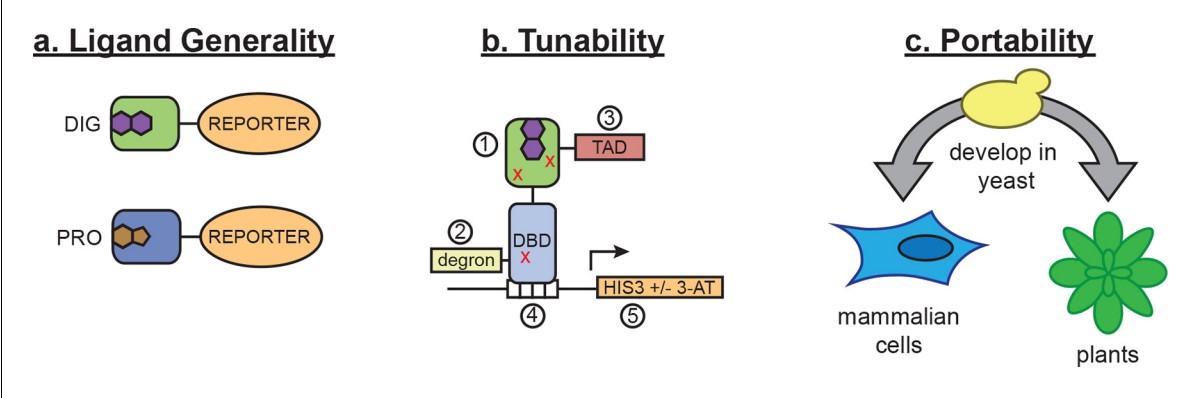

**Figure 7.** Schematic of biosensor platform. (**a**) Biosensors for small molecules are modularly constructed by replacing the LBD with proteins possessing altered substrate preferences. (**b**) Activity of the biosensor can be tuned by 1) introducing destabilizing mutations (red Xs), 2) adding a degron, 3) altering the strength of the TAD or DNA binding affinity of the TF, 4) changes in the number of TF-binding sites or sequences, and 5) titrating 3-aminotriazole, an inhibitor of His3. (**c**) Yeast provide a genetically tractable chassis for biosensor development before implementation in more complex eukaryotes, such as mammalian cells and plants.

(~600 Å$^2$). These protein folds are well suited for *de novo* design of other LBDs because of their large binding pocket and natural substrate diversity (*Todd et al., 2002*). Exploiting dimer interfaces to modulate stability without impairing ligand binding may be a general mechanism to confer conditional stability on LBDs. This possibility is supported by the observation that interface mutations in DIG$_0$ and Gal4 conferring digoxin-dependent stability lead to progesterone-dependent stability in a progesterone biosensor (*Figure 1—figure supplement 2a*).

A long-standing challenge in metabolic engineering is to rapidly detect and control how changes to the regulation and composition of biosynthetic pathways affect product titers. Transcriptional control by a product or intermediate (*Zhang et al., 2012*; *Raman et al., 2014*; *Tang and Cirino, 2011*) and directed evolution of constituent pathway elements (*Agresti et al., 2010*; *Alper et al., 2005*; *Dietrich et al., 2013*) have emerged as promising strategies towards this goal. These approaches require high selectivity against intermediates (*Zhang and Keasling, 2011*), a feature demonstrated here that can be explicitly considered during the computational design and screening process. Our method allows biosensors to be generated that are highly selective for a small molecule, facilitating a simple directed evolution strategy without requiring prior structural or bioinformatic knowledge about the targeted enzyme(s) or pathway(s). Because the biosensors are TF-based, sophisticated systems of optimizing metabolic output, such as dynamic control of gene expression (*Zhang et al., 2012*) and feedback-regulated genome evolution (*Chou and Keasling, 2013*), are possible.

A reliance on the general principles of protein stability and ligand binding allows the development of biosensors that function in any organism with similar protein quality control machinery. Here, we engineered biosensors based on a designed scaffold derived from a bacterial protein. These biosensors required only minimal modifications to retain high levels of sensitivity when developed in yeast and deployed across mammalian and plant species, demonstrating unprecedented portability for biosensors. In some cases, these biosensors showed a greater dynamic range in mammalian cells relative to yeast, possibly due to larger cell volume and variations in protein degradation machinery. Further work using biosensors based on multiple different LBD scaffolds and introduced into diverse organisms will allow us to better understand the principles by which biosensor variability across hosts arises.

Small molecule biosensors with the modularity incorporated here enable diverse cellular responses to a variety of exogenous and endogenous signals (*Banaszynski et al., 2008*). Gene editing is an area that requires particularly tight coupling of cell response to activation signals. The CRISPR/Cas9 system provides a facile and robust genome-editing platform, but it can result in off-target genetic changes (*Fu et al., 2013*; *Mali et al., 2013*; *Pattanayak et al., 2013*). Proposed solutions include optimizing guide RNA sequences (*Fu et al., 2014*; *Cho et al., 2014*), building chimeric

Cas9 fusions requiring the presence of two Cas9 molecules in close proximity (*Mali et al., 2013*; *Tsai et al., 2014*; *Guilinger et al., 2014*; *Ran et al., 2013*), and regulating Cas9 activity by chemical or light-based inducers (*Dow et al., 2015*; *Zetsche et al., 2015*; *Polstein and Gersbach, 2015*). While small molecule inducers including doxycycline and rapamycin have been used, these molecules may confer leaky expression and cytotoxicity (*Xie et al., 2008*). Thus, an expanded chemical repertoire is needed for tightly regulated gene editing and gene therapy applications. By exploiting the low background of the LBD-biosensors, we produced biosensor-Cas9 fusions with tightly controlled activation (*Figure 5e*). This switch-like control over CRISPR/Cas9 activity could reduce background activity and off-target editing, a critical feature for safer gene therapies (*Mandal, et al., 2014*; *Wu et al., 2013*; *Schwank et al., 2013*).

Our biosensor design approach should have numerous applications in agriculture. For example, biosensors could be developed to enable plants to monitor the environment for pollutants, toxins or dangerous compounds. Coupling biosensors with a phytoremediation trait could enable plants to both sense a contaminant and activate a bioremediation gene circuit. When paired with an agronomic or biofuel trait, such biosensors could serve as triggers for bioproduction. In the transgenic *Arabidopsis* plants, we observed ligand-dependent activation in all cells, tissues and organs examined (*Figure 6*).

The technology introduced here operates at either the transcriptional or post-translational level. These biosensors can be developed in yeast and readily transferred with minimal modification to other eukaryotic cell types, where they retain a high level of sensitivity (*Figure 7c*). The generality of our approach arises from the universality of the transcriptional activation and protein degradation machinery across eukaryotes, together with the modularity and tunability of the constituent parts. These biosensors should find broad application, including improving metabolically engineered pathway flux and product titers, exerting ligand-dependent control over genome editing, and detecting exogenous small molecules or endogenous metabolites.

## Materials and methods

### Culture and growth conditions

Biological replicates are defined as samples inoculated from distinct colonies. Growth media consisted of YPAD (10 g/L yeast extract, 20 g/L peptone, 40 mg/L adenine sulfate, 20 g/L glucose) and SD media (1.7 g/L yeast nitrogen base without amino acids, 5 g/L ammonium sulfate, 20 g/L glucose and the appropriate amount of dropout base with amino acids [Clontech, Mountain View, CA]). The following selective agents were used when indicated: G418 (285 mg/L), pen/strep (100 U/mL penicillin and 100 µg/mL streptomycin).

### LBD-yEGFP library construction

The DIG10.3 sequence (*Tinberg et al., 2013*) was cloned by Gibson assembly (*Gibson et al., 2009*) into a pUC19 plasmid containing yeast enhanced GFP (yEGFP, UniProt ID B6UPG7) and a KanMX6 cassette flanked by 1000 and 500 bp upstream and downstream homology to the *HO* locus. The DIG10.3 sequence was randomized by error-prone PCR using a Genemorph II kit from Agilent Technologies. An aliquot containing 100 ng of target DNA (423 bp out of a 7.4 kb plasmid) was mixed with 5 µL of 10X Mutazyme buffer, 1 µL of 40 mM dNTPS, 1.5 µL of 20 µM forward and reverse primer containing 90 bp overlap with the pUC19 plasmid (oJF70 and oJF71), and 1 µL of Mutazyme polymerase in 50 µL. The reaction mixture was subject to 30 cycles with Tm of 60°C and extension time of 1 min. Vector backbone was amplified using Q5 polymerase (NEB, Ipswich, MA) with oJF76 and oJF77 primers with Tm of 65°C and extension time of 350 s. Both PCR products were isolated by 1.5% agarose gel electrophoresis and the randomized target was inserted as a genetic fusion to yEGFP by Gibson assembly (*Gibson et al., 2009*). Assemblies were pooled, washed by ethanol precipitation, and resuspended in 50 µL of dH$_2$O, which was drop dialyzed (EMD Millipore, Billerica, MA) and electroporated into E. cloni supreme cells (Lucigen, Middleton, WI). Sanger sequencing of 16 colonies showed a mutation rate of 0–7 mutations/kb. The library was expanded in culture and maxiprepped (Qiagen, Valencia, CA) to 500 µg/µl aliquots. 16 µg of library was drop dialyzed and electrotransformed into yeast strain Y7092 for homologous recombination into the HO locus.

Integrants were selected by growth on YPAD solid media containing G418 followed by outgrowth in YPAD liquid media containing G418.

## LBD-yEGFP library selections

Libraries of $DIG_0$-yEGFP and $PRO_0$-yEGFP integrated into yeast strain Y7092 were subject to three rounds of fluorescence activated sorting in a BD FACSAria IIu. For the first round, cells were grown overnight to an $OD_{600}$ of ~1.0 in YPAD containing steroid (500 μM digoxigenin or 50 μM progesterone), and cells showing the top 5% of fluorescence activation were collected and expanded overnight to an $OD_{600}$ of ~1.0 in YPAD lacking steroid. In the second sort, cells displaying the lowest ~3% fluorescence activation were collected. Cells passing the second round were passaged overnight in YPAD containing steroid to an $OD_{600}$ of ~1.0 and sorted once more for the upper 5% of fluorescence activation. The sorted libraries were expanded in YPAD liquid culture and plated on solid YPAD media. Ninety-six colonies from each library were clonally isolated and grown overnight in deep well plates containing 500 μL of YPAD. Candidates were diluted 1:50 into two deep well plates with SD-complete media: one plate supplemented with steroid and the other with DMSO vehicle. Cells were grown for another 4 h, and then diluted 1:3 into microtitre plates of 250 μL of the same media. Candidates were screened by analytical flow cytometry on a BD LSRFortessa cell analyzer. The forward scatter, side scatter, and yEGFP fluorescence (530 nm band pass filter) were recorded for a minimum of 20,000 events. FlowJo X software was used to analyze the flow cytometry data. The fold activation was calculated by normalizing mean yEGFP fluorescence activation for each steroid to the mean yEGFP fluorescence in the DMSO only control. Highest induction candidates were subject to Sanger sequencing with primers flanking the LBD sequence.

## TF-biosensor reporter plasmid construction and integration

Reporter genes were cloned into the integrative plasmid pUG6 or the CEN plasmid pRS414 using the Gibson method (*Gibson et al., 2009*). Each reporter (either yEGFP or firefly luciferase) was cloned to include a 5' *GAL1* promoter (*S. cerevisiae GAL1* ORF bases (-455)-(-5)) and a 3' *CYC1* terminator. For integration, linearized PCR cassettes containing both the reporter and an adjacent KanMX antibiotic resistance cassette were generated using primers containing 50 bp flanking sequences of homology to the *URA3* locus. Integrative PCR product was transformed into the yeast strain PJ69-4a using the Gietz method (*Gietz and Schiestl, 2007*) to generate integrated reporter strains.

## G-DIG/PRO-V plasmid construction

G-DIG/PRO-V fusion constructs were prepared using the Gibson method (*Gibson et al., 2009*). Constructs were cloned into the plasmid p416CYC (p16C). Gal4 (residues 1–93, UniProt ID P04386), DIG10.3 (*Tinberg et al., 2013*), and VP16 (residues 363-490, UniProt ID P06492) PCR products for were amplified from their respective templates using Phusion high-fidelity polymerase (NEB, Waltham, MA) and standard PCR conditions (98°C 10 s, 60°C 20 s, 72°C 30 s; 30 cycles). The 8-residue linker sequence GGSGGSGG was used between Gal4 and DIG10.3. PCR primers were purchased from Integrated DNA technologies and contained 24–30 5' bases of homology to either neighboring fragments or plasmid. Clones containing an N-terminal degron were similarly cloned fusing residues 1–67 of Matα2 (UniProt ID P0CY08) to the 5'- end of G-DIG-V. Plasmids were transformed into yeast using the Gietz method (*Gietz and Schiestl, 2007*), with transformants being plated on synthetic complete media lacking uracil (SD -ura).

## G-DIG-V mutant construction

Mutations were introduced into DIG10.3/pETCON (*Tinberg et al., 2013*) or the appropriate G-DIG/PRO-V construct using Kunkel mutagenesis (*Kunkel, 1985*). Oligos were ordered from Integrated DNA Technologies, Inc. For mutants constructed in pETCON/DIG10.3, the mutagenized DIG10.3 gene was amplified by 30 cycles of PCR (98°C 10 s, 61°C 30 s, 72°C 15 s) using Phusion high-fidelity polymerase (NEB) and 5'- and 3'- primers having homologous overlap with the DIG10.3-flanking regions in p16C-G-DIG-VP64 (Gal4_DIG10.3_VP64_hr_fwd and Gal4_DIG10.3_VP64_hr_rev_rc). Genes were inserted into p16C-Gal4-(HE)-VP16 by Gibson assembly (*Gibson et al., 2009*) using vector digested with HindIII and EcoRI-HF.

## G-PRO-V mutant construction

The gene for DIG10.3 Y34F/Y99F/Y101F was amplified from the appropriate DIG10.3/pETCON (*Tinberg et al., 2013*) construct by 30 cycles of PCR (98°C 10 s, 59°C 30 s, 72°C 15 s) using Phusion high-fidelity polymerase (NEB) and 5'- and 3'- primers having homologous overlap with the DIG10.3-flanking regions in p16C-G-DIG-VP64 (DIG_fwd and DIG_rev). Genes were inserted into p16C-GDVP16 by Gibson assembly (*Gibson et al., 2009*) using p16C-Gal4-(HE)-VP16 vector digested with HindIII and EcoRI-HF.

## G-DIG-V error-prone library construction

A randomized G-DIG-V library was constructed by error-prone PCR using a Genemorph II kit from Agilent Technologies (Santa Clara, CA). An aliquot containing 20 ng p16C GDVP16, 20 ng p16C GDVP16 E83V, and 20 ng p16C Y36H was mixed with 5 µL of 10X Mutazyme buffer, 1 µL of 40 mM dNTPS, 1.5 µL of 20 µM forward and reverse primer containing 37- and 42-bp overlap with the p16C vector for homologous recombination, respectively (GDV_ePCR_fwd and GDV_ePRC_rev), and 1 µL of Mutazyme polymerase in 50 µL. The reaction mixture was subjected to 30 cycles of PCR (95°C 30 s, 61°C 30 s, 72°C 80 s). Template plasmid was digested by adding 1 µL of DpnI to the reaction mixture and incubating for 3 hr at 37°C. Resulting PCR product was purified using a Quiagen PCR cleanup kit, and a second round of PCR was used to amplify enough DNA for transformation. Gene product was amplified by combining 100 ng of mutated template DNA with 2.5 µL of 10 µM primers (GDV_ePCR_fwd and GDV_ePRC_rev), 10 µL of 5X Phusion buffer HF, 1.5 µL of DMSO, and 1 µL of Phusion high-fidelity polymerase (NEB, Waltham, MA) in 50 µL. Product was assembled by 30 cycles of PCR (98°C 10 s, 65°C 30 s, 72°C 35 s). Following confirmation of a single band at the correct molecular weight by 1% agarose gel electrophoresis, the PCR product was purified using a Quaigen PCR cleanup kit and eluted in ddH$_2$O. Yeast strain PyE1 ΔPDR5 was transformed with 9 µg of amplified PCR library and 3 µg of p16C Gal4-(HE)-VP16 triply digested with SalI-HF, BamHI-HR, and EcoRI-HF using the method of Benatuil (*Benatuil et al., 2010*), yielding ~10$^6$ transformants. Following transformation, cells were grown in 150 mL of SD -ura media. Sanger sequencing of 12 individual colonies revealed an error rate of ~1–6 mutations per gene.

## G-DIG-V library selections

An error-prone library of G-DIG$_0$/DIG$_1$/DIG$_2$/-V transformed into yeast strain PyE1 ΔPDR5 was subjected to three rounds of cell sorting using a Cytopeia (BD Influx) fluorescence activated cell sorter. For the first round, cells displaying high fluorescence in the presence of digoxin (on-state) were collected. Transformed cells were pelleted by centrifugation (4 min, 4000 rpm) and resuspended to a final OD$_{600}$ of 0.1 in 50 mL of SD -ura media, pen/step antibiotics, and 5 µM digoxin prepared as a 100 mM solution in DMSO. The library was incubated at 30°C for 9 hr and then sorted. Cells displaying the highest fluorescent values in the GFP channel were collected (1,747,058 cells collected of 32,067,013 analyzed; 5.5%), grown up at 30°C in SD -ura, and passaged twice before the next sort. For the second round of sorting, cells displaying low fluorescence in the absence of digoxin (off-state) were collected. Cells were pelleted by centrifugation (4 min, 4000 rpm) and resuspended to a final OD$_{600}$ of 0.1 in 50 mL of SD -ura media supplemented with pen/strep antibiotics. The library was incubated at 30°C for 8 hr and then sorted. Cells displaying low fluorescent values in the GFP channel were collected (1,849,137 cells collected of 22,290,327 analyzed; 11.1%), grown up at 30°C in SD -ura, and passaged twice before the next sort. For the last sorting round, cells displaying high fluorescence in the presence of digoxin (on-state) were collected. Cells were prepared as for the first sort. Cells displaying the highest fluorescent values in the GFP channel were collected (359,485 cells collected of 31,615,121 analyzed; 1.1%). After the third sort, a portion of cells were plated and grown at 30°C. Plasmids from 12 individual colonies were harvested using a Zymoprep Yeast miniprep II kit (Zymo Research Corporation, Irvine, CA) and the gene was amplified by 30 cycles of PCR (98°C 10 s, 52°C 30 s, 72°C 40 s) using Phusion high-fidelity polymerase (NEB) with the T3 and T7 primers. Sanger sequencing (Genewiz, Inc., South Plainfield, NJ) was used to sequence each clone in the forward (T3) and reverse (T7) directions.

## G-DIG-V error-prone library mutation screens

Of 12 sequenced clones from the library sorts, two showed significantly improved (>2-fold) response to DIG over the input clones (clone 3 and clone 6). Clone 3 contains the following mutations: Gal4_T44T (silent), Gal4_L77F, DIG10.3_E5D, DIG10.3_E83V, DIG10.3_R108R (silent), DIG10.3_L128P, DIG10.3_I137N, DIG10.3_S143G, and VP16_A44T. Clone 6 contains the following mutations: Gal4_R60S, Gal4_L84L (silent), VP16_G17G (silent), VP16_L48V, and VP16_H98H (silent). To identify which mutations led to the observed changes in DIG response, variants of these clones with no silent mutations and each individual point mutant were constructed using Kunkel mutagenesis (*Kunkel, 1985*). Oligos were ordered from Integrated DNA Technologies, Inc. Sequence-confirmed plasmids were transformed into PyE1 ΔPDR5f and plated onto selective SD -ura media. Individual colonies were inoculated into liquid media, grown at 30°C, and passaged once. Cells were pelleted by centrifugation (4 min, 1700 × g) and resuspended to a final $OD_{660}$ of 0.1 in 1 mL of SD -ura media supplemented 50 µM DIG prepared as a 100 mM solution in DMSO. Following a 6 hr incubation at 30°C, cells were pelleted, resuspended in 200 µL of PBS, and cellular fluorescence was measured on an Accuri C6 flow cytometer using a 488 nm laser for excitation and a 575 nm band pass filter for emission. FlowJo software version 7.6 was used to analyze the flow cytometry data. The data are given as the mean yEGFP fluorescence of the single yeast population in the absence of DIG (off-state) and the mean yEGFP fluorescence of the higher fluorescing yeast population in the presence of DIG (on-state).

## Computational model of Gal4-DIG

A model of the Gal4-DIG10.3 fusion was built using Rosetta Remodel (*Huang et al., 2011*) to assess whether the linker between Gal4 and the DIG LBD, which are both dimers, would allow for the formation of a dimer in the fusion construct. In the simulation, the Gal4 dimer was held fixed while the relative orientation of the DIG LBD monomers were sampled symmetrically using fragment insertion in the linker region. Constraints were added across the DIG LBD dimer interface to facilitate sampling. The lowest energy model satisfied the dimer constraints, indicating that a homodimer configuration of the fusion is possible.

## TF-biosensor titration assays in yeast

Yeast strain PyE1 transformed with p16C plasmids containing G-LBD-V variants were inoculated from colonies into SD –ura media supplemented and grown at 30°C overnight (16 h). 10 µL of the culture was resuspended into 490 µL of separately prepared media each containing a steroid of interest (SD –ura media supplemented the steroid of interest and DMSO to a final concentration of 1% DMSO). Resuspended cultures were then incubated at 30°C for 8 hr. 125 µL of incubated culture was resuspended into 150 µL of fresh SD –ura media supplemented with the steroid of interest and DMSO to a final concentration of 1%. These cultures were then assayed by analytical flow cytometry on a BD LSRFortessa using a 488 nm laser for excitation. The forward scatter, side scatter, and yEGFP fluorescence (530 nm band pass filter) were recorded for a minimum of 20,000 events. FlowJo X software was used to analyze the flow cytometry data. The fold activation was calculated by normalizing mean yEGFP fluorescence activation for each steroid to the mean yEGFP fluorescence in the DMSO only control. G-PRO$_0$-V was assayed on a separate day from the other TF biosensors under identical conditions.

## TF-biosensor kinetic assays in yeast

Yeast strain PyE1 was transformed with p16C plasmids containing G-LBD-V variants were inoculated from colonies into SD –ura media and grown at 30°C overnight (16 hr). 5 µL of each strain was diluted into 490 µL of SD –ura media in 2.2 mL plates. Cells were incubated at 30°C for 8 hr. 5 µL of steroid was then added for a final concentration of 250 µM digoxin or 50 µM progesterone. For each time point, strains were diluted 1:3 into microtitre plates of 250 µL of the same media. Strains were screened by analytical flow cytometry on a BD LSRFortessa cell analyzer. The forward scatter, side scatter, and yEGFP fluorescence (530 nm band pass filter) were recorded for a minimum of 20,000 events. FlowJo X software was used to analyze the flow cytometry data. The fold activation was calculated by normalizing mean yEGFP fluorescence activation for each time point to the mean yEGFP fluorescence at T = 0 hr.

## Luciferase reporter assay

Yeast strains containing either a plasmid-borne or integrated luciferase reporter were transformed with p16C plasmids encoding TF-biosensors. Transformants were grown in triplicate overnight at 30°C in SD –ura media containing 2% glucose in sterile glass test tubes on a roller drum. After ~16 hr of growth, $OD_{600}$ of each sample was measured and cultures were back diluted to $OD_{600}$ = 0.2 in fresh SD –ura media containing steroid dissolved in DMSO or a DMSO control (1% DMSO final). Cultures were grown at 30°C on roller drum for 8 hr prior to taking readings. Measurement of luciferase activity was adapted from a previously reported protocol(Leskinen et al., 2003). 100 µL of each culture was transferred to a 96-well white NUNC plate. 100 µL of 2 mM D-luciferin in 0.1 M sodium citrate (pH 4.5) was added to each well of the plate and luminescence was measured on a Victor 3V after 5 min.

## Yeast deletion strain creation

Genomic deletions were introduced into the yeast strains PJ69-4a and PyE1 using the 50:50 method (Horecka and Davis, 2014). Briefly, forward and reverse primers were used to amplify an URA3 cassette by PCR. These primers generated a product containing two 50 bp sequences homologous to the 5' and 3' ends of the ORF at one end and a single 50 bp sequence homologous to the middle of the ORF at the other end. PCR products were transformed into yeast using the Gietz method (Gietz and Schiestl, 2007) and integrants were selected on SD –ura plates. After integration at the correct locus was confirmed by a PCR screen, single integrants were grown for 2 days in YEP containing 2.5% ethanol and 2% glycerol. Each culture was plated on synthetic complete plates containing 5-fluoroorotic acid. Colonies were screened for deletion of the ORF and elimination of the Ura3 cassette by PCR and confirmed by Sanger sequencing.

## TF-biosensor specificity assays

Yeast strains expressing the TF-biosensors and yEGFP reporter (either genetically fused or able to be transcriptionally activated by the TAD) were grown overnight at 30°C in SD –ura media for 12 hr. Following overnight growth, cells were pelleted by centrifugation (5 min, 5250 rpm) and resuspended into 500 µL of SD –ura. 10 µL of the washed culture was resuspended into 490 µL of separately prepared media each containing a steroid of interest (SD –ura media supplemented with the steroid of interest and DMSO to a final concentration of 1% DMSO). Steroids were tested at a concentration of 100 µM digoxin, 50 µM progesterone, 250 µM pregnenolone, 100 µM digitoxigenin, 100 µM beta-estradiol, and 100 µM hydrocortisone. Stock solutions of steroids were prepared as a 50 mM solution in DMSO.

Resuspended cultures were then incubated at 30°C for 8 hr. 125 µL of incubated culture was resuspended into 150 µL of fresh SD –ura media supplemented the steroid of interest, and DMSO to a final concentration of 1%. These cultures were then assayed by analytical flow cytometry on a BD LSRFortessa using a 488 nm laser for excitation. The forward scatter, side scatter, and yEGFP fluorescence (530 nm band pass filter) were recorded for a minimum of 20,000 events. FlowJo X software was used to analyze the flow cytometry data. The fold induction was calculated by normalizing mean yEGFP fluorescence activation for each steroid to the mean yEGFP fluorescence in the DMSO only control.

## Yeast spotting assays

Yeast strain PJ69-4a transformed with p16C plasmids containing degron-G-DIG-V variants were first inoculated from colonies into SD -ura media and grown at 30°C overnight (16 hr). 1 mL of each culture was pelleted by centrifugation (3000 rcf, 2 min), resuspended in 1 mL of fresh SD -ura and the $OD_{660}$ was measured. Each culture was then diluted in SD -ura media to an $OD_{660}$ = 0.2 and incubated at 30°C for 4–6 hr. 1 mL of each culture was pelleted and resuspended in sterile, distilled water and the $OD_{660}$ measured again. Each transformant was then diluted to an $OD_{660}$ = 0.1. Four 1/10 serial dilutions of each culture were prepared in sterile water (for a total of 5 solutions). 10 µL of each dilution was spotted in series onto several SD –ura –his agar plates containing 1 mM 3-aminotriazole and the indicated steroid. Steroid solutions were added to agar from 200x steroid solutions in DMSO (0.5% DMSO final in plates).

## 3β-HSD plasmid and library construction

The 3β-HSD ORF was synthesized as double-stranded DNA (Integrated DNA Technologies, Inc., Coralville, IA) and amplified using primers oJF325 and oJF326 using KAPA HiFi under standard PCR conditions and digested with BsmBI to create plasmid pJF57. 3β-HSD expression plasmids (pJF76 through pJF87) were generated by digesting plasmid pJF57 along with corresponding plasmids from the Yeast Cloning Toolkit (Lee et al., 2015) with BsaI and assembled using the Golden Gate Assembly method (Engler et al., 2008). The 3β-HSD sequence was randomized by error-prone PCR using a Genemorph II kit from Agilent Technologies. An aliquot containing 100 ng of target DNA was mixed with 5 µL of 10X Mutazyme buffer, 1 µL of 40 mM dNTPS, 1.5 µL of 20 µM forward and reverse primer containing 90-bp overlap with the 3β-HSD expression plasmids and 1 µL of Mutazyme polymerase in 50 µL. The reaction mixture was subject to 30 cycles with Tm of 60°C and extension time of 1 min. Vector backbone was amplified using KAPA HiFi polymerase with oJF387 and oJF389 (pPAB1) or oJF387 and oJF389 (pPOP6) with Tm of 65°C and extension time of 350 s. PCR products were isolated by 1.5% agarose gel electrophoresis and assembled using the Gibson method (Gibson et al., 2009). Assemblies were pooled, washed by ethanol precipitation, and resuspended in 50 µL of dH$_2$O, which was drop dialyzed (Millipore) and electroporated into E. cloni supreme cells (Lucigen). Sanger sequencing of 16 colonies showed a mutation rate of 0–4 mutations/kb. The library was expanded in culture and maxiprepped (Qiagen) to 500 µg/µL aliquots. 16 µg of library was drop dialyzed and electrotransformed into yeast strain PyE1.

## 3β-HSD progesterone selections

PyE1 transformed with libraries of 3β-HSD were seeded into 5 mL of SD –ura –leu media supplemented and grown at 30°C overnight (24 hr). Cultures were measured for OD$_{600}$, diluted to an OD$_{600}$ of 0.0032, and 100 µL was plated onto SD –ura –leu –his plates supplemented 35 mM 3-AT and either 50 µM pregnenolone or 0.5% DMSO.

## Progesterone bioproduction and GC/MS analysis

Production strains were inoculated from colonies into 5 mL SD –ura media and grown at 30°C overnight (16 hr). 1 mL of each culture was washed and resuspended into 50 mL of SD –ura with 250 µM of pregnenolone and grown at 30°C for 76 hr. OD$_{600}$ measurements were recorded for each culture before pelleting by centrifugation. Cells were lysed by glass bead disruption, and lysates and growth media were extracted separately with heptane. Extractions were analyzed by GC/MS.

## TF-biosensor EGFP assays in mammalian cells

The K562 cell line was obtained from the ATCC. The cell line was not authenticated and not tested for mycoplasma contamination. For each TF-biosensor, 1 µg of the PiggyBac construct along with 400 ng of transposase were nucleofected into K562 cells using the Lonza Nucleofection system as per manufacturer settings. Two days post-transfection, cells underwent puromycin selection (2 µg/mL) for at least eight additional days to allow for unintegrated plasmid to dilute out and ensure that all cells contained the integrated construct. An aliquot of 100,000 cells of each integrated population were then cultured with 25 µM of progesterone, 1 µM of digoxigenin, or no small molecule. Forty-eight hours after small molecule addition, cells were analyzed by flow cytometry using a BD Biosciences Fortessa system. Mean EGFP fluorescence of the populations was compared.

## Construction of K562 cell lines

The PiggyBac transposase system was employed to integrate biosensor constructs into K562 cells. Vector PB713B-1 (Systems Biosciences, Mountain View, CA) was used a backbone. Briefly, this backbone was digested with NotI and HpaI and G-LBD-V, Gal4BS-E1b-EGFP (EGFP; enhanced GFP ref or UniProt ID A0A076FL24), and sEF1-Puromycin were cloned in. Gal4BS represents four copies of the binding sequence. For hCas9, the PiggyBac system was also employed, but the biosensors were directly fused on the N-terminus of Cas9 and were under the control of the CAGGS promoter. Cas9 from *S. pyogenes* was used.

## TF-biosensor-Cas9 assays

Construct integration was carried out as for the Cas9 experiments for EGFP assays, except that the constructs were integrated into K562 containing a broken EGFP reporter construct. Introduction of an engineered nuclease along with a donor oligonucleotide can correct the EGFP and produce fluorescent cells. Upon successful integration (~10 days after initial transfection), 500,000 cells were nucleofected with 500 ng of guide RNA (sgRNA) and 2 µg of donor oligonucleotide. Nucleofected cells were then collected with 200 µL of media and 50 µL aliquots were added to wells containing 950 µL of media. Each nucleofection was split into four separate wells containing 1 µM of digoxigenin, 25 µM of progesterone, or no small molecule. Forty-eight hours later, cells were analyzed using flow cytometry and the percentage of EGFP positive cells was determined.

## TF-biosensor assays in protoplasts

Digoxin transcriptional activators were initially tested in a transient expression assay using *Arabidopsis* protoplasts according previously described methods(*Yoo et al., 2007*), with some modifications. Briefly, protoplasts were prepared from 6-week old *Arabidopsis* leaves excised from plants grown in short days. Cellulase Onozuka R-10 and Macerozyme R-10 (Yakult Honsha, Inc., Japan) in buffered solution were used to remove the cell wall. After two washes in W5 solution, protoplasts were re-suspended in MMg solution at $2 \times 10^5$ cells/mL for transformation. Approximately $10^4$ protoplasts were mixed with 5 µg of plasmid DNA and PEG4000 at a final concentration of 20%, and allowed to incubate at room temperature for 30 min. The transformation reaction was stopped by the addition of two volumes of W5 solution, and after centrifugation, protoplasts were re-suspended in 200 µL of WI solution (at $5 \times 10^5$/mL) and plated in a 96-well plate. Digoxigenin (Sigma-Aldrich, St. Louis, MO) was added to the wells, and protoplasts were incubated overnight at room temperature in the dark, with slight shaking (40 rpm). For luciferase imaging, protoplasts were lysed using Passive Lysis Buffer (Promega, Madison, WI) and mixed with LARII substrate (Dual-Luciferase Reporter Assay System, Promega, Madison, WI). Luciferase luminescence was collected by a Stanford Photonics XR/MEGA-10 ICCD Camera and quantified using Piper Control (v.2.6.17) software.

## Plant plasmid construction

G-DIG$_1$-V was recoded to function as a ligand-dependent transcriptional activator in plants. Specifically, an *Arabidopsis thaliana* codon optimized protein degradation sequence from the yeast MAT$\alpha$2 gene was fused in frame in between the Gal4 DBD and the DIG$_1$ LBD. The resulting gene sequence was codon-optimized for optimal expression in *Arabidopsis thaliana* plants and cloned downstream of a plant-functional CaMV35S promoter to drive constitutive expression in plants, and upstream of the octopine synthase (*ocs*) transcriptional terminator sequence. To quantify the transcriptional activation function of DIG10.3, the luciferase gene from *Photinus pyralis* (firefly) was placed downstream of a synthetic plant promoter consisting of five tandem copies of a Gal4 Upstream Activating Sequence (UAS) fused to the minimal (-46) CaMV35S promoter sequence. Transcription of luciferase is terminated by the E9 terminator sequence. These sequences were cloned into a pJ204 plasmid and used for transient expression assays in Arabidopsis protoplasts.

## Construction of transgenic Arabidopsis plants

After confirmation of function in transient tests, the digoxin biosensor genetic circuit was transferred to pCAMBIA 2300 and was stably transformed into *Arabidopsis thaliana* ecotype Columbia plants using a standard Agrobacterium floral dip method (*Clough and Bent, 1998*). Transgenic plants were selected in MS media (*Murashige and Skoog, 1962*) containing 100 mg/L kanamycin.

## TF-biosensor assays in transgenic plants

Transgenic plants expressing the digoxin biosensor genetic circuit were tested for digoxigenin-induced luciferase expression by placing 14–16 day-old plants in liquid MS (- sucrose) media supplemented with 0.1 mM digoxigenin in 24-well plates, and incubated in a growth chamber at 24°C, 100 $\mu E.m^2.s^{-1}$ light. Luciferase expression was measured by imaging plants with a Stanford Photonics XR/MEGA-10 ICCD Camera, after spraying luciferin and dark adapting plants for 30 min. Luciferase expression was quantified using Piper Control (v.2.6.17) software. Plants from line KJM58-10 were used to test for the specificity of induction by incubating plants, as described above, in 0.1 mM

digoxigenin, 0.1 mM digitoxigenin, and 0.02 mM β-estradiol. All chemicals were obtained from Sigma-Aldrich (St. Louis, MO).

## Acknowledgements

We thank James DiCarlo and Caleb Bashor for advice on construct development and strain engineering. This work was supported by grants from the Defense Threat Reduction Agency (DTRA) to DB and JIM and NIH grant P41 GM103533 to the UW Yeast Resource Center supporting DB and SF GMC received support from the Department of Energy Grant DE-FG02-02ER63445 and from the Synthetic Biology Engineering Research Center (SynBERC) through awards MCB-134189 and EEC-0540879 from the National Science Foundation. JF is a National Science Foundation Graduate Research Fellow (DGE1144152) and DJM is a Howard Hughes Medical Institute Fellow of the Life Sciences Research Foundation. DB and SF are investigators of the Howard Hughes Medical Institute.

## Additional information

### Competing interests

JF, BWJ, CET, DJM, RC, XR, GMC, SF, DB: Harvard University has filed a provisional patent on this work. The other authors declare that no competing interests exist.

### Funding

| Funder | Grant reference number | Author |
|---|---|---|
| National Science Foundation | Graduate Research Fellowship DGE1144152 | Justin Feng |
| Howard Hughes Medical Institute | Life Sciences Research Foundation Fellowship | Daniel J Mandell |
| Defense Threat Reduction Agency | | June I Medford David Baker |
| U.S. Department of Energy | DE-FG02- 02ER63445 | George M Church |
| National Science Foundation | Synthetic Biology Engineering Research Center (SynBERC) through awards MCB-134189 and EEC-0540879 | George M Church |
| National Institutes of Health | P41 GM103533 | Stanley Fields David Baker |
| Howard Hughes Medical Institute | | Stanley Fields David Baker |

The funders had no role in study design, data collection and interpretation, or the decision to submit the work for publication.

### Author contributions

JF, BWJ, CET, DJM, Conception and design, Acquisition of data, Analysis and interpretation of data, Drafting or revising the article; MSA, RC, KJM, Acquisition of data, Analysis and interpretation of data, Drafting or revising the article; XR, Established the Cas9 GFP correction K562 cell line, Contributed unpublished essential data or reagents; JIM, GMC, SF, DB, Conception and design, Analysis and interpretation of data, Drafting or revising the article

## Additional files

### Supplementary files

• Supplementary file 1. Oligonucleotides used in the construction of biosensors are shown.

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
