## [Decision Letter]

Thank you for submitting your work entitled "A general strategy to construct small molecule biosensors in eukaryotes" for consideration by *eLife*. Your article has been reviewed by two peer reviewers, and the evaluation has been overseen by Jeffery Kelly (Reviewing editor) and John Kuriyan (Senior Editor).

The reviewers have discussed the reviews with one another and the Reviewing editor has drafted this decision to help you prepare a revised submission.

Please consider all the reviewer comments and suggestions pasted below and address as many of them as possible in the two month time frame.

The authors are requested to spend a few sentences addressing whether their current strategy lends itself to quantification, or whether the use of coupled, intermolecular systems (i.e., transcription of a reporter gene) precludes accurate quantification. The idea that these biosensors can be quickly ported from yeast to other eukaryotes is compelling and important, but the authors did not rigorously support this claim with experiments specifically designed to show and characterize organism-to-organism portability. Addressing this would be desirable.

The editors feel that the paper could benefit from a careful reading, with an eye towards integration of the results and conclusions, done by one of the senior authors.

*Reviewer #1:*

This is a nice paper from the Church, Fields, and Baker labs. I think that many will find this paper interesting and I recommend it for publication with one suggestion (not a requirement) for additional data.

Responding to the appearance of the analyte appears to be the primary goal of this work. However, biosensors of this sort would be extremely useful for single-cell analyses of biological processes such as signaling (think of Tsien's calcium dyes). In this light, it is important for the reader to know how these engineered biosensors respond to the loss of the analyte as well as the appearance of the analyte, which is well-treated in the current paper. I imagine the authors have already done these experiments, and analyte withdrawal data would nicely complement the data currently in Figure 2 (as new panels G and H, showing a time course of signal loss when dig or prog are removed).

One definition of "biosensor" only requires detection of a particular analyte. A more ambitious definition would capture quantification of the analyte. I would appreciate if the authors could spend a few sentences addressing whether their current strategy lends itself to quantification, or whether the use of coupled, intermolecular systems (i.e., transcription of a reporter gene) precludes accurate quantification. This same issue likely prevents these biosensors from responding rapidly/accurately to the disappearance of analyte. This is fine. I don't think the authors should be asked to invent version 2 of the system for this paper, but I do think it would be a good idea for the authors to address this issue in the text. Having protein stability as the readout makes a biosensor less sensitive (no amplification from transcription), but it also makes it more responsive to analyte in terms of speed and quantification.

One issue missing from this study is how the presence of the biosensor might change or affect the behavior of the cell. In other words, progesterone that is sequestered by the biosensor is not available to bind to the endogenous target. Additionally, this biosensor strategy is "general" (Figure 7) only insofar as a researcher has access to a high-affinity, protein-based ligand-binding domain for a particular analyte.

*Reviewer #2:*

This work is impressive in the scope of different design features/perturbations and applications implemented in engineering biosensors. However, most of these features have been previously reported (and appropriately cited in each case by the authors), and it is therefore difficult to determine the novelty and utility for the greater scientific community. If the authors' intended finding is an overall approach of implementing many different strategies to achieve an improved end goal application, and that their strategy is generalizable, more efficient, or possessing of some other hitherto lacking feature, then the manuscript would need to more directly demonstrate this. As one example, the idea that these biosensors can be quickly ported from yeast to other eukaryotes is compelling and important, but the authors did not rigorously support this claim with experiments specifically designed to show and characterize organism-organism portability. That is, transferring a set of proteins and showing they work in other organisms may not be a sufficiently systematic study that statistically supports organism-organism portability. If the authors could better refine/clarify the contribution of their work, this would help to motivate readers to adopt the strategies and findings from this work.

The authors showed tremendous versatility in demonstrating several potential applications. However, the depth of each of these demonstrations was somewhat lacking and appeared more as minimal proof of principle experiments. For example, the authors demonstrated increased bioproduction of progesterone. This could indeed serve as a compelling model of how to engineer and use biosensors. However, the effects seen were not large (~2 fold with large error bars). Perhaps if any one of the applications were flushed out in more depth, more detail, or with greater effect, this would have greater impact on the community.

---

## [Author Response]

Reviewer #1:

*Responding to the appearance of the analyte appears to be the primary goal of this work. However, biosensors of this sort would be extremely useful for single-cell analyses of biological processes such as signaling (think of Tsien's calcium dyes). In this light, it is important for the reader to know how these engineered biosensors respond to the loss of the analyte as well as the appearance of the analyte, which is well-treated in the current paper. I imagine the authors have already done these experiments, and analyte withdrawal data would nicely complement the data currently in Figure 2 (as new panels G and H, showing a time course of signal loss when dig or prog are removed).*

The reviewer raises an important point. We have added new panels G and H to Figure 2 discussion to the last paragraph of the subsection “TF-biosensors amplify ligand-dependent responses” characterizing the behavior of TF-biosensors when withdrawing ligand after activation. We observe rapid decrease in fluorescence levels with 50% reduction after approximately 5 hours, and almost complete loss after 10-15 hours. Destabilization likely occurs more rapidly than observed by fluorescence, as the reduction in fluorescence signal is dependent on both the degradation of the TF-biosensors after ligand withdrawal, as well as the degradation and dilution of previously expressed yEGFP.

*One definition of "biosensor" only requires detection of a particular analyte. A more ambitious definition would capture quantification of the analyte. I would appreciate if the authors could spend a few sentences addressing whether their current strategy lends itself to quantification, or whether the use of coupled, intermolecular systems (i.e., transcription of a reporter gene) precludes accurate quantification. This same issue likely prevents these biosensors from responding rapidly/accurately to the disappearance of analyte. This is fine. I don't think the authors should be asked to invent version 2 of the system for this paper, but I do think it would be a good idea for the authors to address this issue in the text. Having protein stability as the readout makes a biosensor less sensitive (no amplification from transcription), but it also makes it more responsive to analyte in terms of speed and quantification.*

We thank the reviewer for pointing out the need to distinguish between signal from bulk populations versus single cells. The levels of accuracy and noise at single-cell resolution are important aspects when deciding whether our biosensors in both the direct fusion or the TF systems are suitable for different applications. We have added single cell flow cytometry event data as Figure 1—figure supplement 1 and Figure 2—figure supplement 2, together with a discussion in the subsection “TF-biosensors amplify ligand-dependent responses”. We observe a more rapid response with direct fusions at the expense of dynamic range. We also observe tighter distributions using direct fusions, which would allow for more accurate quantification of ligand (Figure 1—figure supplement 1 and Figure 2—figure supplement 2). At very high levels of ligand, the TF-biosensors exhibit tight distributions as well, but at intermediate ligand concentrations, the GFP levels span a fairly broad range. We used FACS to isolate cells from the dark population peak and found those cells to be inviable. Further optimizations could possibly reduce noise in the TF system, such as integrating the constructs instead of using a plasmid-borne system, or exploring a larger library of promoters.

*One issue missing from this study is how the presence of the biosensor might change or affect the behavior of the cell. In other words, progesterone that is sequestered by the biosensor is not available to bind to the endogenous target. Additionally, this biosensor strategy is "general" (Figure 7) only insofar as a researcher has access to a high-affinity, protein-based ligand-binding domain for a particular analyte.*

The reviewer correctly points out the importance of developing biosensors with activity that is compatible with intended downstream applications. In this study, we chose two steroids orthogonal to the yeast sterol biosynthesis pathway in order to reduce biological responses from the host. 3β-HSD mutations identified in the histidine selections and yEGFP activation assays were re-transformed into yeast strains not expressing our TF-biosensors for quantification using GC/MS, so the improvements in 3β-HSD bioproduction were observed both with and without the progesterone biosensors present.

The CYC1 promoter used to express the TF-biosensors leads to fairly low expression and causes only a small amount of ligand to be sequestered. If an application were particularly sensitive to very minute changes in ligand concentration, the TF-biosensor could be used with an even weaker promoter.

It is also true that this method requires a high-affinity ligand-binding domain as a starting point, but nearly any small molecule of interest will have a natural protein that interacts with it. Using de novodesigned binders opens the possibility of generating biosensors for ligands with unsuitable or unknown binding proteins.

We have added text to the Discussion to discuss these key points.

Reviewer #2:

*This work is impressive in the scope of different design features/perturbations and applications implemented in engineering biosensors. However, most of these features have been previously reported (and appropriately cited in each case by the authors), and it is therefore difficult to determine the novelty and utility for the greater scientific community. If the authors' intended finding is an overall approach of implementing many different strategies to achieve an improved end goal application, and that their strategy is generalizable, more efficient, or possessing of some other hitherto lacking feature, then the manuscript would need to more directly demonstrate this. As one example, the idea that these biosensors can be quickly ported from yeast to other eukaryotes is compelling and important, but the authors did not rigorously support this claim with experiments specifically designed to show and characterize organism-organism portability. That is, transferring a set of proteins and showing they work in other organisms may not be a sufficiently systematic study that statistically supports organism-organism portability. If the authors could better refine/clarify the contribution of their work, this would help to motivate readers to adopt the strategies and findings from this work.*

We thank the reviewer for suggesting further discussion on the key issue of multi-organism portability of our biosensors. Previous work has shown that conditionally destabilized proteins are degraded by the highly conserved ubiquitin proteasome system (Egeler et al., 2011). Relying on the general principles of protein stability and ligand binding allows for the development of biosensors in any organism with similar protein quality control machinery. The fact that our biosensors engineered from a bacterial protein scaffold retained high levels of sensitivity with minimal modifications when developed in yeast and deployed across mammalian and plant species demonstrates unprecedented portability for biosensors and suggests than even further portability may be possible. We have added text to the fourth paragraph of the Discussion to clarify these points.

We do agree that further work on portability will allow us to better understand the principles for using biosensors across diverse organisms, which we believe will require multiple novel biosensors originating from multiple LBD scaffolds, forming the basis of an intriguing follow-up study.

The authors showed tremendous versatility in demonstrating several potential applications. However, the depth of each of these demonstrations was somewhat lacking and appeared more as minimal proof of principle experiments. For example, the authors demonstrated increased bioproduction of progesterone. This could indeed serve as a compelling model of how to engineer and use biosensors. However, the effects seen were not large (~2 fold with large error bars). Perhaps if any one of the applications were flushed out in more depth, more detail, or with greater effect, this would have greater impact on the community.

The GC/MS results show statistically significant increases in progesterone bioproduction when compared to wildtype after a 76 hr period. The difference in the TF-biosensor yEGFP activation assay and the GC/MS results may be due to a difference in the experimental length, with the pregnenolone possibly becoming limited over the course of a three-day experiment. In addition, the identified mutations were discovered after a single round of random mutagenesis with no previous knowledge of the enzyme’s structure or function applied. We believe a more thorough search of the sequence space, either through saturation mutagenesis or iterative rounds of selections and error-prone PCR, would yield other beneficial mutations.